# Optimization Mechanism of Nozzle Parameters and Characterization of Nanofibers in Centrifugal Spinning

**DOI:** 10.3390/nano13233057

**Published:** 2023-11-30

**Authors:** Qinghua Guo, Peiyan Ye, Zhiming Zhang, Qiao Xu

**Affiliations:** 1Hubei Digital Textile Equipment Key Laboratory, Wuhan Textile University, Wuhan 430200, China; 2115373039@mail.wtu.edu.cn (Q.G.); 2115373044@mail.wtu.edu.cn (P.Y.); 2School of Mechanical Engineering and Automation, Wuhan Textile University, Wuhan 430200, China; 2006109@wtu.edu.cn

**Keywords:** nanofiber, centrifugal spinning, optimization, bent-tube nozzle

## Abstract

Nanofibers are an emerging kind of nano-material, widely used in several application domains such as biomedicine, high-efficiency filtration media, precision electronics, and optical devices. Centrifugal spinning, which is a novel nanofiber production technology, has been widely studied. This paper proposes a structural parameter optimization design method of a bent-tube nozzle. The mathematical model of the spinning solution motion in the nozzle is first developed. The optimization function of the structure parameters of the bent-tube nozzle is then obtained by calculation. Afterwards, these parameters are optimized using a neural network algorithm. The obtained results show that, when the bending angle is 15°, the curvature radius is 10 mm, the outlet radius is 0.205 mm, and the head loss of the solution can be minimized. Finally, centrifugal spinning experiments are conducted and the influence of the centrifugal spinning parameters on the nanofibers is analyzed. In addition, the optimized bent-tube nozzle improves the surface morphology of the nanofibers as their diameter distribution becomes more uniform.

## 1. Introduction

Nanofibers are one type of fiber materials with a diameter of tens to hundreds of nanometers [1,2]. Materials composed of nanofibers have high performance, such as high porosity, outstanding mechanical properties, and a good face area ratio [3,4]. The main characteristics of nanofiber materials are the macroscopic quantum tunneling effect, small size effect, surface effect, and quantum size effect [5]. Due to these unique characteristics, nanofibers are widely used in human tissue architecture, high-performance filters, biofilm materials, flat media, batteries, biological products, and reinforced composites [6,7,8]. There are many traditional methods used to produce nanofibers, such as electrostatic spinning [9,10], melt blowing [11], phase separation [12], etc. In electrostatic spinning, a high-voltage electric field is applied to the solution in the container, and the droplet at the nozzle changes from a ball shape to a Taylor cone shape under the effect of the electric field. The tip of the Taylor cone will be stretched, and the solution behind the Taylor cone becomes thinner and longer, gradually forming fibers [13]. In melt-blown spinning, polymer particles are heated at high temperature and extruded by the equipment to the capillary nozzle. Driven by high-speed gas on both sides, they are gradually stretched and elongated, mixed with the surrounding gas, and, finally, cooled into fibers [11]. However, traditional nanofiber preparation methods show some limitations. In electrostatic spinning, the solution needs to be applied with a high-voltage electric field [8,14]. In melt-blown spinning, the production efficiency is low, the cost is high, and melt-blown spinning needs to be carried out at high temperature, which results in poor morphological properties of the fibers [15].

Hence, high-speed centrifugal spinning [16,17,18,19] has been proposed. Compared with traditional nanofiber preparation methods, high-speed centrifugal spinning has strengths with a wide range of raw material sources, and nanofibers can be prepared at room temperature and normal pressure without applying a high-voltage electric field [20]. Furthermore, high-speed centrifugal spinning is energy-efficient and environmentally friendly [21]. The quality of nanofibers is affected by internal factors (e.g., the solution concentration and solute type [18,22], etc.) and external factors (e.g., the motor speed, collection distance, nozzle diameter, and nozzle structure [23,24,25]). Nanofibers made from one raw material are referred to as single fibers, and nanofibers made from multiple raw materials are referred to as composite fibers [26].

The diameter distribution and morphology of nanofibers were researched by changing the nozzle diameter [27]. The influence of the nozzle direction on the initial jet motion was studied [28]. Four different types of nozzles were proposed, and researchers proved which one was the most suitable nozzle for centrifugal spinning [29].

For the purpose of obtaining nanofibers with a better surface quality and morphology, the structural parameters of the nozzles should be optimized. In this paper, the minimum energy loss in the nozzle is used as the optimization objective function, and a neural network algorithm [30,31] is used to optimize the nozzle structure to obtain the optimal combination of parameters for the nozzles. Thinking, in neural network algorithms, refers to the process of reasoning based on logical rules. First, the neural network algorithm turns information into a concept, and use symbols to express it. Second, according to a symbol operation, it carries out logical reasoning in serial mode [32].

This paper further examines the flow rules of a solution in the container and nozzles, analyzes the force on the solution inside the nozzle, and optimizes the nozzle structure parameters based on neural network algorithms. The force of the solution in the container is first analyzed based on the laws of energy and mass conservation, and the mathematical model of the solution in the nozzle is developed. The parameters of the nozzle structure are then optimized using the neural network algorithm. Afterwards, centrifugal spinning experiments are conducted to verify the optimization results. The morphology and surface quality of the nanofibers are tested by scanning electron microscopy (SEM).

## 2. Solution Flow Model in Nozzle

In order to further study high-speed centrifugal spinning, the mathematical model of the spinning solution flow in the container and nozzle was developed. Through the force analysis of the spinning solution movement in the nozzle, the force balance formula was obtained. The solution model was deduced from the conservation of energy and mass, and the flow law of the spinning solution in the container was obtained.

### 2.1. Principle of Centrifugal Spinning and Jet Process

The centrifugal spinning equipment is shown in Figure 1. The centrifugal spinning equipment mainly includes a driving motor, transmission device, safety net, a container, two nozzles, and fiber collector.

The motor first rotates under the drive of the frequency conversion controller, and the container will be driven to rotate through the transmission shaft. When the motor speed reaches a critical value, the spinning solution in the container then flows to the nozzles at both ends of the container under the action of centrifugal force. With the increase of the motor speed, the centrifugal force on the spinning solution gradually increases. When the centrifugal force reaches a certain critical value, it becomes equal to the sum of the viscous force and surface force. After that, the speed of the motor continues to increase and the spinning solution will be ejected from the nozzles under the action of centrifugal force by overcoming the viscous force and surface force of the solution, whereupon the solution is thinned and stretched in the air. The solvent evaporates after that and forms nanofibers. Finally, nanofibers are collected on the fiber collection device.

### 2.2. Establishment of Spinning Solution Flow Model in Nozzle

To simplify the model, the container and nozzles are facilitated to a two-dimensional plane figure as shown in Figure 2A. It can be clearly seen from Figure 2A that the container part in the figure is left- and right-symmetrical except for the nozzle. The bending direction of the bent-tube nozzle part is just opposite, and the nozzle size is completely consistent. For design purposes, the right half of Figure 2A is taken to develop the motion model, as shown in Figure 2B.

In the Figure 2B, a rectangular co-ordinate system x-o-y is established, where the *x*-axis coincides with the tank axis, and the co-ordinate origin is at the intersection of the tank axis and the tank centerline. Starting from the origin, any length along the positive direction of the *x*-axis is *L*. The diameter of the container is *2R*_1_. *L*_1_ is the length from the rotation axis to the conical transition zone, and this zone is denoted by Zone *I*. *L*_2_ is the length of the conical transition zone along the *x*-axis direction, and this area is denoted by Zone *II*. *L*_3_ is the straight pipe part of the nozzle, and this area is denoted by *III*. The bent-tube nozzle is denoted by Zone *IV*. The transition angle of Zone *II* is *φ*. The curvature radius of the bent tube is *R*. The outlet diameter is *2R*_2_ and the bending angle of the bent-tube nozzle is *Θ*.

The stress analysis of the solution in area *I* is shown in Figure 3A. In the horizontal direction of the solution storage tank, when the centrifugal spinning equipment is working, the solution is mainly subjected to the viscous force *F_μ_*. The direction is negative along the *x*-axis. The centrifugal force is *F_cen_* and the direction is positive along the *x*-axis. To facilitate the analysis, the flow state of the spinning solution in Zone *I* is considered as the laminar flow, and the velocity distribution is shown in Figure 3B.

It can be seen from Figure 3A that the solution motion equation is as follows:(1)mdVdt=Fcen−Fμ

When the equipment runs stably, Formula (1) can be written as follows:(2)mdVdt=mLω2−k(dudr)2

In Formula (2), *m* is the mass of the solution in Zone *I*, *V* is the average velocity of the solution in Zone *I*, *L* is any length from the center of rotation to the conical transition zone, *ω* refers to the rotation angular velocity of the tank body after the smooth operation of the equipment, and *k* is the viscosity coefficient of the solution.

Let *L* = *L*_1_. From Formula (2), the velocity expression *V_I_* of the solution at the junction of Zone *I* and Zone *II* can be obtained:(3)VI=mL1ω2−k(dudt)nmt0

*t*_0_ is the working time of the spinning equipment, and *n* is the rheological index of the solution. Because of the viscosity, the motion of the spinning solution will be blocked, and friction will be formed between the moving fluids and between the fluid and the solid wall surface. Therefore, that part of the moving mechanical energy will be converted into heat energy. The head loss of the spinning solution along the way in Zone *I* is:(4)hfI=λL12R1V22g

In the above formula, *L*_1_ is the length of Zone *I*, 2*R*_1_ is the tank diameter, *V* represents the average speed *V*_1_ of Zone *I*, and *g* is the acceleration of gravity. *λ* is the loss coefficient along the way, which is determined by the following formula:(5)λ=64Re

*R_e_* is the Reynolds number which is a dimensionless constant used to characterize the ratio of inertial force to viscous force when the spinning solution flows. In this paper, the Reynolds number is calculated as:(6)Re=ρVdμ

In the above formula, *ρ* is the solution density. *V* is the average spinning solution speed in the container, which is equal to *V*_1_ in value. *d* is numerically equal to 2*R*_1_, and *μ* is the dynamic viscosity of the spinning solution.

Introduce Formulae (5) and (6) into Formula (4) to obtain the head loss along the path in Zone *I*, which is as follows:(7)hfI=8μL1VIρRI2g

The stress analysis of the solution in Zone *II* is carried out, as shown in Figure 3C. The spinning solution is mainly subjected to centrifugal force *F_cen_* in Zone *II* and the viscous force of the spinning fluid itself *F_μ_*. The resistance *F_ω_* is exerted on the solution by the vessel wall in the conical transition Zone *II*. The wall is symmetrical up and down so the components of *F_ω_* along the y-axis direction cancel each other out and the direction of the combined force is along the negative direction of the *x*-axis. When considering the viscous effect of the solution, due to the reduction of the conical transition area from *A*_1_ to *A*_2_, the spinning solution will form a vortex area in the conical transition area and cause local kinetic energy loss as shown in Figure 3D.

In order to facilitate the study, the viscous effect of the solution is ignored in the stress analysis here. The force acting on the wall of the conical transition zone is:(8)Fw=ρVII2d2A1(A1−A2)2

In Formula (8), *A*_1_ is the area of the entrance side of the conical transition zone. *A*_2_ is the area of the exit side. *ρ* is the density of the spinning solution, and *V_II_* is the average speed of the spinning solution in the conical transition zone. It can be seen from the above formula that, when *A*_1_ is equal to *A*_2_ and the viscosity of the spinning solution is ignored, the wall of the conical transition zone does not work on the spinning solution. According to the law of conservation of mass, the mass of the spinning solution flowing into and out of Zone *II* in unit time is equal, as seen in the following formula:(9)−∫A1ρ1vn1dA=∫A2ρ2vn2dA

In Formula (9), *v_n_*_1_ is the external normal velocity of the solution at the inlet, and *v_n_*_2_ is the external normal velocity of the solution at the outlet. Here, *V_I_* and *V_II_* are used to replace them, respectively. In the process of centrifugal spinning, the density of the spinning solution remains unchanged, so:(10)ρ1=ρ2

Take Formula (10) into Formula (9) and simplify Formula (9) to obtain the expression of *V_II_*:(11)VII=VIA1A2

In the actual production process of nanofibers, the spinning solution has a certain viscosity. Friction is formed between the solution and the inner wall of the nozzles. As the solution flows through the conical region, it creates a swirling motion, which dissipates some of the mechanical energy. The local head loss dissipated when the spinning solution passes through the conical transition zone is as follows:(12)hjII=ξVII22g
where *V_II_* is the exit velocity of Zone *II*, and *ξ* is the local head loss coefficient. The calculation formula of *ξ* is as follows:(13)ξ=l(1ε−1)2+λ8tan(φ/2)(1−A22A12)

*ε* is the intermediate quantity introduced for the convenience of calculation. Its value is determined by the inlet and outlet area of the intermediate transition zone. The value of the coefficient *l* is related to the magnitude of *φ*. When the value of *φ* is 90 °, the value of *l* is 0.35. The specific calculation method is as follows:(14)ε=0.57+0.0431.1−(A2/A1)

Take Formulae (13) and (14) into (12), and the local head loss in Zone *II* is:(15)hjII=[l(10.57+0.0431.1−(A2/A1)−1)2+λ8tan(φ/2)(1−A22A12)]VII22g

The stress analysis of solution in Zone *III* is carried out, as shown in Figure 3E. For the convenience of research, the flow state of the spinning solution in Zone *III* is considered as the laminar flow, and the velocity distribution is shown in Figure 3F.

According to the law of conservation of mass:(16)VIII=VII

The head loss of the spinning solution along the path in Zone *III* is:(17)hfIII=8μL3VIIIρR22g

The stress analysis of the solution in Zone *IV* is shown in Figure 3G. The spinning solution is mainly subjected to the centrifugal force *F_cen_* in Zone *IV*, the viscous force *F_μ_*, and the force exerted by the nozzle wall on the solution *F_W_*. In the bent part of the nozzle, due to the presence of the bend, the bent part will change the velocity direction and pressure distribution of the spinning solution. Thus, the bent tube will create a flow separation at the bend, creating a vortex area between the solution and the wall, as shown in Figure 3H.

In order to study the magnitude of the force *F_W_*, the cross-sectional area of the bent-tube nozzle is set as *A*. According to the momentum equation:(18)ρQ(VIV→−VIII→)=F→−(P1−Pa)n1→A−(P2−Pa)n2→A

In Formula (18), n1→ and n2→ are the unit vectors of the external normal directions at the inlet and outlet of the bent tube, respectively; VIII→ and VIV→ are the velocity vectors at the inlet and outlet; (P1−Pa) and (P2−Pa) are the relative pressures at the inlet and outlet of the bent tube, respectively; and *Q* is the volumetric flow rate. The vector equation is written into the component form as follows:(19){Fx=ρAVIII2(cosθ−1)+(P2−Pa)Acosθ−(P1−Pa)AFy=ρAVIII2sinθ+(P2−Pa)Asinθ

According to the force composition theorem, the force exerted by the bent-tube wall on the spinning solution is obtained as follows:(20)Fw=([ρAVIII2(cosθ−1)+(P2−Pa)Acosθ−(P1−Pa)A]2+[ρAVIII2sinθ+(P2−Pa)Asinθ]2)12

According to the law of conservation of mass and the law of velocity composition, the outlet velocity of the bent-tube nozzle is:(21)VIV=VIIIcosθ

*Θ* is the bending angle of the bent-tube nozzle. The Formulae (3), (11), and (16) are introduced into Formula (21), and the specific expression of the outlet velocity of the bent-tube nozzle is as follows:(22)VIV=[mL1ω2−k(dudr)n]R12cosθm[R1−L2tan(φ/2)]2t0

Because the spinning solution is viscous in the centrifugal spinning process, the viscosity of the solution will block the spinning solution jet process, which will consume a certain amount of mechanical energy of movement. Friction forms between the solution and the wall of the bent-tube nozzle, which converts part of the mechanical energy into heat energy. The local head loss of the solution in the Zone *IV* is:(23)hjIV=ξVIV22g
where *V_IV_* is the average velocity of the solution in the bent tube. *ξ* is the local head loss coefficient, which is determined by the following formula:(24)ξ=[0.131+0.163(2R2R)72]2θπ

*Θ* is the bending angle of the bent tube, *R* is the radius of curvature, and 2*R*_2_ is the outlet diameter of the bent-tube nozzle.

## 3. Optimization Process of Bent-Tube Nozzle Structure

The head loss during the centrifugal spinning process should be minimized to obtain nanofibers with a more uniform diameter distribution and better surface morphology. In other words, in the spinning process, when the solution passes through the nozzle, the energy loss should be reduced.

### 3.1. Establishment of Optimization Model of Centrifugal Spinning Nozzle

The main structural parameters of spinning container are: *R*_1_, *L*_1_, *L*_2_, *L*_3_, *φ*, *Θ*, *R*, and *R*_2_. Considering the influence of the equipment parameters and solution parameters on the experiment in the actual centrifugal spinning experiment, *Θ*, *R*, and *R*_2_ were chosen as the optimized design variables. During centrifugal spinning, the total energy loss of the solution in the container is given by:(25)E=hfI+hjII+hfIII+hjIV
where *E* is the total energy loss of the solution in the container during centrifugal spinning. The following optimization objective function is established according to the minimization of the energy loss during spinning:(26)max f(x)=min E(x)

The design variables of the model can be expressed as:(27)X={bending angleΘradius of curvatureRnozzle radiusR2}T

The determination of the parameter range of the spinning equipment and solution will improve the optimization efficiency of the structural parameters of the centrifugal spinning nozzle. The parameter range of the centrifugal spinning system is shown in Table 1.

The simplified fitting function can be written as follows:(28)f(x)=−8.2×10−4(R2+R22)sin2θ−29.5θ90−39.1

### 3.2. Application of Neural Network Algorithm in Bent-Tube Nozzle

Neural network algorithms’ logical thinking is the process of reasoning according to logical rules. It first turns information into a concept and represents them symbolically. Then, according to a symbol operation, it carries out logical reasoning in serial mode. This process can be written into serial instructions for the computer to execute. The basic points of this way of thinking are the following two things: information is stored on the network through a distribution of excitation patterns on neurons, and information processing is performed through a dynamic process of simultaneous interactions between neurons. The artificial neural network does not need to determine the mathematical equation of the mapping relationship between the input and output in advance. It only learns some rules through its own training and obtains the result closest to the expected output value when the input value is given.

The number of intermediate layers and the number of neurons in each layer of the network can be arbitrarily set according to the specific situation, and its performance varies with the difference of the structure. The three-layer neural network structure is shown in Figure 4A, where:

The input layer vector is:(29)X=(θ,R,R2)

The hidden layer vector is:(30)H=(f(θ),f(R),f(R2))

The output layer vector is:(31)Y=f(x)

The global error function is:(32)e=12p∑i=1p∑k=1n(dk(i)−fk(i))2
where *p* is the number of samples, *d_k_* is the expected output of the *k^th^* sample, *f_k_* is the actual output of the *k^th^* sample, and *n* is the number of training outputs of a single sample.

The flow chart of neural network algorithm is shown in Figure 4B.

In Figure 4B, *w*_1_ is the weight function between the input layer and the hidden layer. *w*_2_ is the weight function between the hidden layer and the output layer. *b*_1_ and *b*_2_ are offset functions, respectively.

From Figure 5, it can be found that the structural parameters are non-linear and monotonic for the head loss, with a negative z-axis indicating the lost head.

When the head loss is minimized (i.e., the value of the fitting function is maximized), a larger radius of curvature and an appropriate nozzle radius and bending angle are required. The outlet radius of the nozzles and head loss roughly show a positive correlation link. Thus, the head loss decreases with the increase of the outlet radius. A too-small outlet radius will increase the flow field resistance of the spinning solution flow. In addition, the head loss increases with the increase of the bending angle. Considering that the air field generated by the rotation of the equipment perturbs the fibers at the nozzle outlet during the nanofiber production process, the nozzle outlet must have a certain bending angle towards the opposite direction of rotation.

When the number of optimizations is almost 15, the solution energy loss is basically stable and in dynamic equilibrium, as shown in Figure 6. A smaller bending angle, a smaller outlet radius, and a larger radius of curvature can minimize the energy loss during the nanofiber production.

### 3.3. Optimization Results of Bent-Tube Nozzle

It has been demonstrated that a bending angle of 15°, a radius of curvature of 10 mm, and an outlet radius of 0.205 mm present the best combination of the bent nozzle structural parameters.

## 4. Experiment Sections

### 4.1. Centrifugal Spinning Experiment

#### 4.1.1. Experimental Materials and Experimental Equipment

Centrifugal spinning experiments were carried out to verify the accuracy of the optimization results. There are many polymer compounds that can be applied to the preparation of nanofibers by centrifugal spinning—for example, polyethylene oxide (PEO), average molecular weight: 1 × 10^6^, completely soluble in water. PEO exhibits excellent properties such as being non-toxic and non-irritating, and having a low solution rheology, combined action with organic solvents, low ash content, and good thermoplasticity. It is widely used in construction cement mortar, cosmetics, agriculture, food, petroleum, pulp and paper, and other fields. Polyvinyl alcohol (PVA) resin aqueous solution has good film-forming and emulsifying properties. It can be used as a protective colloid during the emulsification and suspension polymerization of polymers. PVA can be used in textile slurry, Vinnie fiber, paper coating agent, building material, adhesive, PVA film, food, and medicine. PVA can also be used as a soil conditioner, polymerization suspension agent, emulsifier, quenching agent, and so on. In addition, with the continuous development and improvement of PVA performance, its use is constantly expanding. Homemade centrifugal spinning equipment and two kinds of nozzles were used in this experiment. The equipment is shown in Figure 7. Centrifugal spinning equipment is used to produce nanofibers.

#### 4.1.2. Experimental Process of Centrifugal Spinning

The preparation of the solution used for the centrifugal spinning experiment consists of several steps. PEO powder was first added to deionized pure water, and the solution was continuously stirred for 8 h using a magnetic stirrer in order to prepare an aqueous PEO solution having a mass concentration of 4%. PVA powder was added to pure water at a temperature of 80 °C, and the solution was continuously stirred for 6 h in order to prepare aqueous PVA solution having a mass concentration of 8%. In order to obtain better nanofibers, the two solutions were separately prepared, and they were added to the centrifugal spinning equipment in a volume ratio of 8:2 to conduct the hybrid centrifugal spinning experiment. The experiment was conducted at an ambient temperature of 24 °C and ambient relative humidity of 42%. Two types of nozzles were used: a straight tube having a nozzle diameter of 0.41 mm (the control group) and a bent tube having a nozzle radius of 0.205 mm, a bending angle of 15°, and a curvature radius of 10 mm (the experimental group). The maximum motor speed can reach 6000 rpm with a frequency converter. The centrifugal force overcomes other external forces and the solutions were ejected out from the nozzles. Then, the solvent was evaporated. Finally, the nanofibers were collected by the nanofiber collection device.

#### 4.1.3. Experimental Results and Nanofiber Diameter Distribution

Samples of the collected nanofibers were extracted for the SEM analysis, allowing us to observe their morphology and surface quality. The SEM images of nanofibers prepared with different nozzles are shown in Figure 8.

In the centrifugal spinning experiment, the flow, gravity, temperature, and humidity fields have a certain impact on the experimental results. It can be seen from Figure 8A that the surface quality of the nanofibers produced with the straight-tube nozzle is large, it is bumpy and uneven, and it has uneven thickness. There are signs of fracture and the droplets are attached to the surface of the fibers. It can also be seen from the diameter distribution diagram that the diameter of nanofibers is mainly distributed in the range of 800–1100 nm, and the diameter distribution is scattered. In conclusion, the nanofibers produced with straight-tube nozzles have poor overall morphology and low surface quality. However, the nanofibers produced with bent-tube nozzles are uniform in thickness, smooth in surface, and free from fracture and liquid drop. In addition, it can be seen from the diameter distribution diagram that the diameter of nanofibers is mainly distributed in the range of 600–900 nm and the overall diameter distribution is relatively centralized. In summary, the nanofibers produced with the optimized bent-tube nozzle have a smaller diameter, more centralized distribution, high surface quality, and very good overall morphology.

### 4.2. The Influence of Spinning Parameters on Nanofibers

#### 4.2.1. Effect of Nozzle Diameter on Nanofibers

The diameter of nanofibers is influenced by the diameter of the bent-tube nozzle. In order to describe the relationship between the diameter of the bent-tube nozzle and that of the nanofibers, bent-tube nozzles with different diameters are used for producing nanofibers. The SEM images of nanofibers are shown in Figure 9. The average diameter of the nanofibers was measured and a line graph was obtained between the nanofiber diameter and the diameter of the bent-tube nozzle, as shown in Figure 10.

It can be seen from Figure 10 that the diameter of the nanofibers is positively correlated with the nozzle diameter. Due to the viscosity of the solution, when the nozzle diameter is too small, the nanofibers will fracture. When the nozzle diameter is too large, the solution is thrown out by centrifugal force before forming fibers at the outlet. Therefore, the optimal value for the nozzle outlet diameter is considered to be 0.41 mm.

#### 4.2.2. Effect of Bending Angle on Nanofibers

The quality of the nanofibers can be changed by varying the bending angle of the nozzles. In the centrifugal spinning process, when the bending angle is 0°, due to the presence of Coriolis force, the solution at the nozzle outlet will experience a shear phenomenon. In order to avoid this phenomenon, the nozzle is optimized to be curved. The SEM images of nanofibers produced by nozzles with different bending angles are shown in Figure 11.

It can be seen that, when the bending angle is 0°, the thickness of the nanofibers is uneven, and there are liquid droplets and fractures on the surface of the nanofibers. When the bending angle is 15°, the surface quality of the nanofibers is good, the thickness is uniform, and there are no droplets or fractures. However, when the bending angle is 30° or 45°, the nanofibers start to stick together and show signs of fracture.

#### 4.2.3. Effect of Curvature Radius on Nanofibers

When the bending angle and nozzle diameter are determined, the curvature radius of the nozzle will only affect the distance traveled by the solution inside the nozzle. In other words, the curvature radius will affect the energy loss of the solution. The SEM images of nanofibers produced with different curvature radii are shown in Figure 12. The line graph of the average diameter of the nanofibers and the curvature radius of the bent-tube nozzle is shown in Figure 13.

It can be seen from Figure 12 and Figure 13 that, when the curvature radius of the bent-tube nozzle is 10 mm, the surface of the nanofibers is smooth, the diameter thickness is uniform, and the average diameter of the nanofibers is the smallest.

## 5. Conclusions

This paper studied the flow of the spinning solution in the container and nozzles, and analyzed the stress and head loss of the solution. The flow model and optimization model of the solution were developed. A neural network algorithm was used to optimize the structural parameters of the nozzle. A centrifugal spinning experiment was conducted to verify the optimization results. The influence of the spinning parameters on the nanofibers was discussed. It was deduced that the head loss of the solution can be minimized when the nozzle bending angle is 15°, the curvature radius is 10 mm, and the nozzle outlet radius is 0.205 mm. The nanofibers produced with optimized nozzles have better surface morphology and a more uniform diameter. However, the influence of gravity and Coriolis force of the solution are ignored, which make the results not satisfactory. Therefore, in future work, an accurate and quantitative analysis of the solution flow will be conducted through fluid simulation.

## Figures and Tables

**Figure 1 nanomaterials-13-03057-f001:**
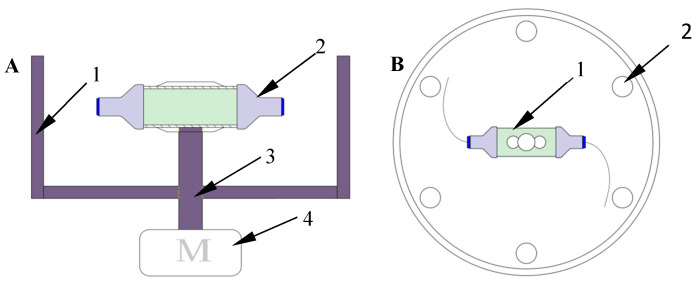
Schematic diagram of centrifugal spinning equipment: (**A**) front view: 1, protection cover; 2, nozzle; 3, transmission device; 4, motor; and (**B**) top view: 1, container; 2, fiber collection rod.

**Figure 2 nanomaterials-13-03057-f002:**
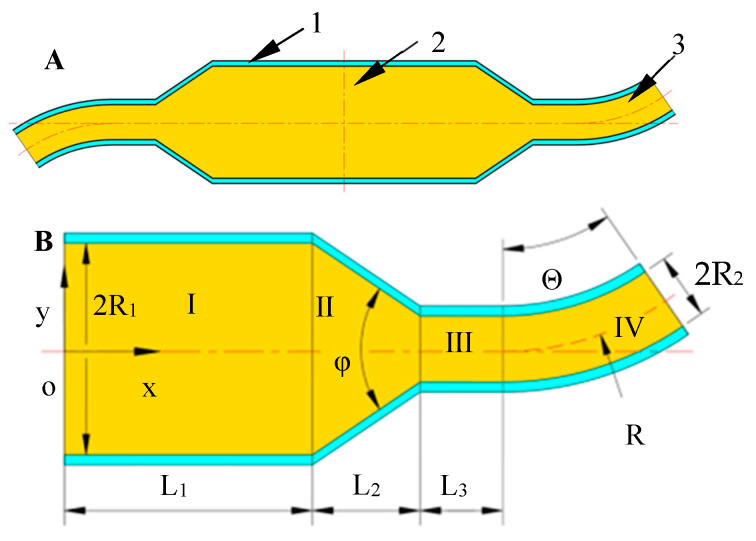
Two-dimensional plan of container and detail labels: (**A**) 1, container; 2, spinning solution; 3, nozzle. (**B**) Analysis of the right half of container and nozzle.

**Figure 3 nanomaterials-13-03057-f003:**
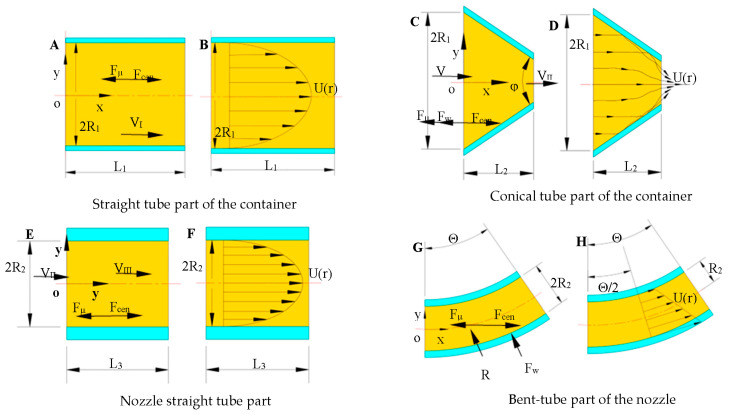
Analysis of the force and velocity distribution in each part of the container and nozzle. (**A**,**C**,**E**,**G**) show the force analysis; (**B**,**D**,**F**,**H**) show the velocity distri-bution.

**Figure 4 nanomaterials-13-03057-f004:**
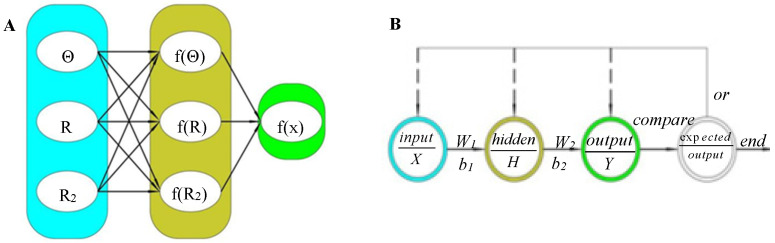
Structure and flow chart of neural network algorithm. (**A**) Neural network structure. (**B**) Flow chart of neural network algorithm.

**Figure 5 nanomaterials-13-03057-f005:**
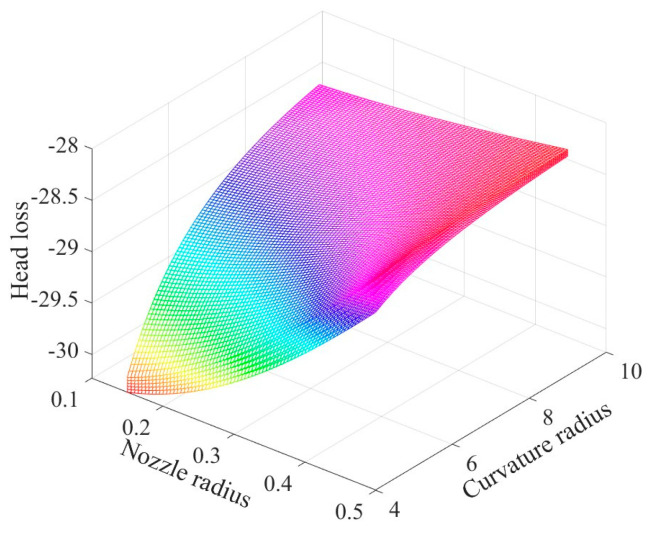
Fitting function diagram based on neural network algorithm.

**Figure 6 nanomaterials-13-03057-f006:**
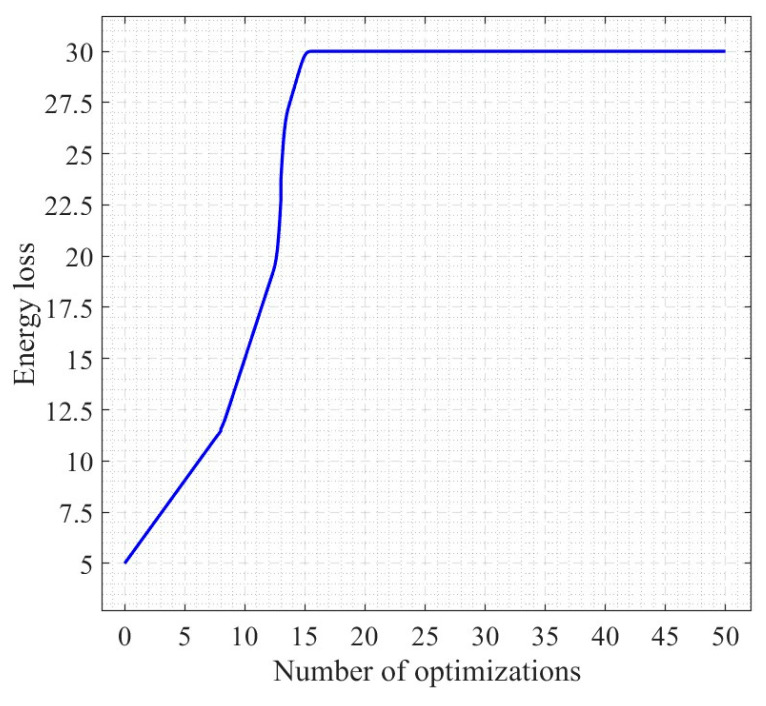
Process of computation of neural network algorithm.

**Figure 7 nanomaterials-13-03057-f007:**
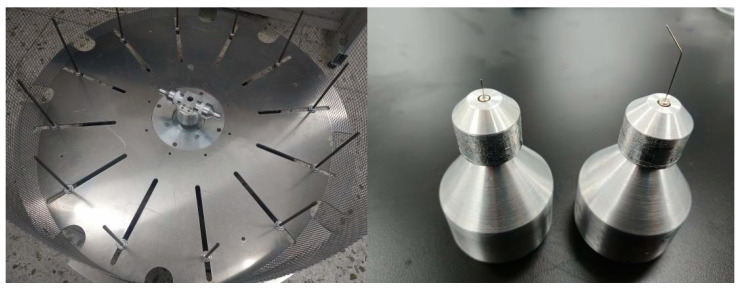
Centrifugal spinning equipment and nozzle.

**Figure 8 nanomaterials-13-03057-f008:**
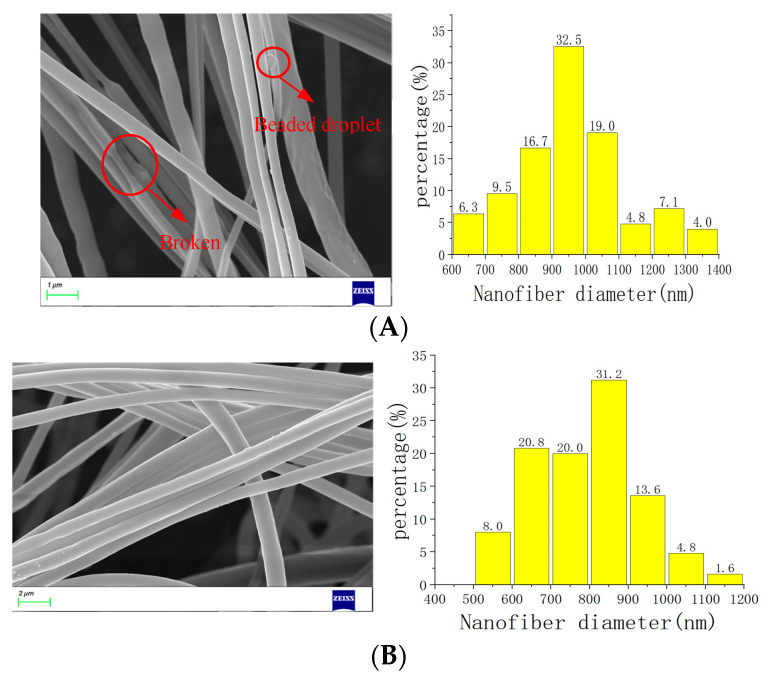
SEM and diameter distribution. (**A**) SEM and diameter distribution diagram of nanofibers in straight-tube nozzle. (**B**) SEM and diameter distribution diagram of nanofibers in bent-tube nozzle.

**Figure 9 nanomaterials-13-03057-f009:**
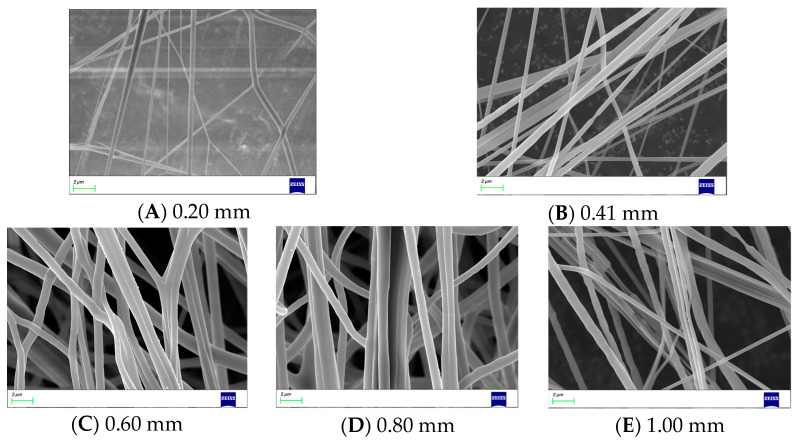
Nanofibers produced with different bent-tube nozzle diameters.

**Figure 10 nanomaterials-13-03057-f010:**
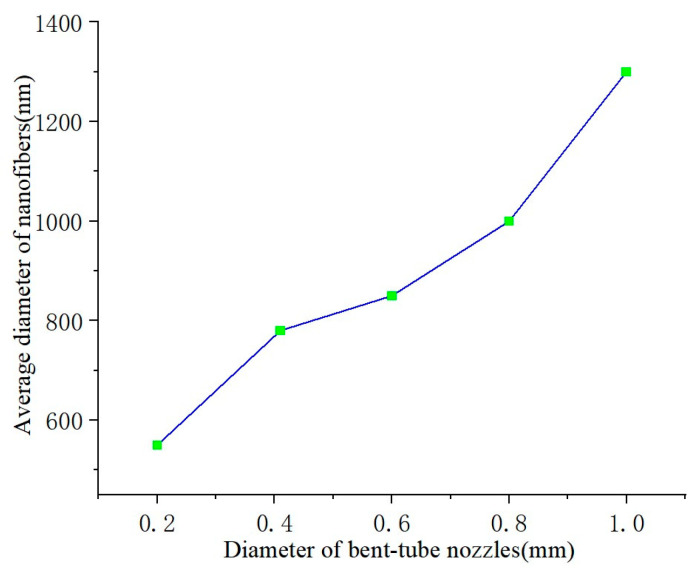
Average diameter of nanofibers.

**Figure 11 nanomaterials-13-03057-f011:**
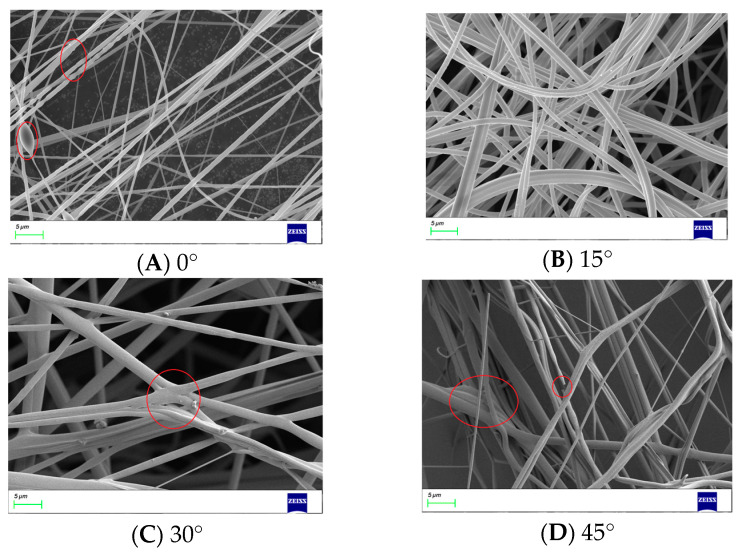
Nanofibers produced by bent-tube nozzles with different bending angles.

**Figure 12 nanomaterials-13-03057-f012:**
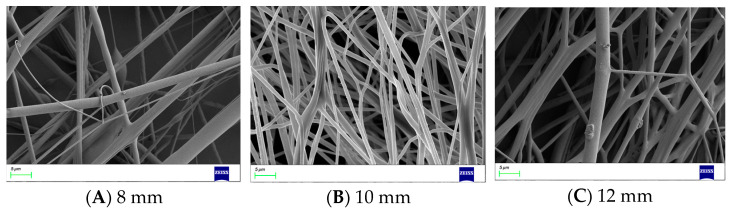
Nanofibers produced with different curvature radii.

**Figure 13 nanomaterials-13-03057-f013:**
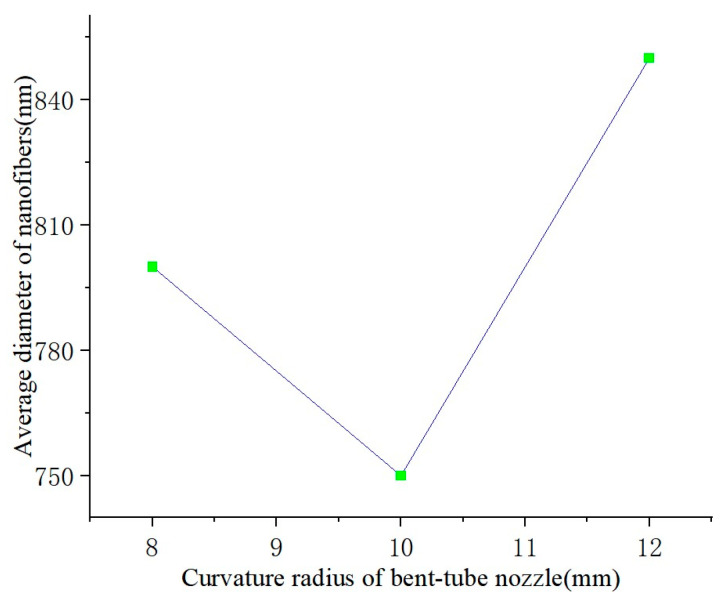
Average diameter of nanofibers.

**Table 1 nanomaterials-13-03057-t001:** Parameters of centrifugal spinning system.

Item	Interpretation	Sign	Value (Range)
Optimization objective	Head loss	*E*	Minimum/j
Design parameters	Bending angle	*Θ*	[0, 45]/angle
	Radius of curvature	*R*	[4, 9]/10^−3^ m
	Outlet radius	*R*_2_	[0.15, 0.5]/10^−3^ m
Equipment parameters	Container length	*L*_1_	30/10^−3^ m
	Taper tube length	*L*_2_	14.4/10^−3^ m
	Length of straight pipe	*L*_3_	10/10^−3^ m
	Container radius	*R*_1_	15/10^−3^ m
	Container taper	*φ*	90/angle
Solution parameters	Consistency coefficient	*k*	7.62/pa·s
	Rheological index	*n*	0.504
	Solution density	*ρ*	1000/kg/m^3^
Speed	Equipment speed	*ω*	[1000, 4000]/rpm

## Data Availability

Data is contained within the article.

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
