# Peer review of "Optimization Mechanism of Nozzle Parameters and Characterization of Nanofibers in Centrifugal Spinning"

_nanomaterials, 2023, doi:10.3390/nano13233057_

Round 1
Reviewer 1 Report (Previous Reviewer 2)
Comments and Suggestions for Authors
The authors took into account a number of previous comments, but the manuscript still contains a large number of typos, blots and errors. Authors should carefully check the text of the manuscript and correct any shortcomings. In some places, the authors miss the dimension. Figures 9 and 10 indicate the dimension mm, but we are talking about nanofibers?! I recommend technical editing before publishing your manuscript.
Comments on the Quality of English LanguageThe text of the manuscript is not neatly formatted.
Author Response
Response to Reviewer 1 Comments
Comments and Suggestions for Authors:
The authors took into account a number of previous comments, but the manuscript still contains a large number of typos, blots and errors. Authors should carefully check the text of the manuscript and correct any shortcomings. In some places, the authors miss the dimension. Figures 9 and 10 indicate the dimension mm, but we are talking about nanofibers?! I recommend technical editing before publishing your manuscript.
Response: Thank you very much for your comments on the manuscript. We have accepted all comments and made revisions. And the modified part is marked red. The manuscript has been polished by a professional polishing agency. In addition, in Figure 9, A 0.20mm, B 0.41mm, C 0.60mm, D 0.80mm, and E 1.00mm respectively represent the outlet diameters of the bent-tube nozzles used. In line chart 10, the horizontal axis represents the outlet diameter of the bent-tube nozzles, and the vertical axis represents the average diameter of the nanofibers produced by the corresponding bent-tube nozzles.
Comments on the Quality of English Language:
The text of the manuscript is not neatly formatted.
Response: The manuscript has been polished by a professional polishing agency, and the Certificate of Language Editing is attached.

Reviewer 2 Report (New Reviewer)
Comments and Suggestions for Authors
The article entitled "Optimization Mechanism of Nozzle Parameters and Characterization of Nanofibers in Centrifugal Spinning" seems to be an extension of the article published by the authors earlier (https://doi.org/10.3389/fbioe.2022.884316). The above article is not cited in the present manuscript. Also, the authors may clearly distinguish the new work from the previous work as both works are identical.
The conclusion lacks the most relevant findings of the study.
The citing of articles in the body of the manuscript is not properly done and "Error! 41 Reference source not found" repeatedly appeared in the manuscript.
Comments on the Quality of English Language
The language is slightly confusing and may need revision.
Author Response
Response to Reviewer 2 Comments
Comments and Suggestions for Authors:
The article entitled "Optimization Mechanism of Nozzle Parameters and Characterization of Nanofibers in Centrifugal Spinning" seems to be an extension of the article published by the authors earlier (https://doi.org/10.3389/fbioe.2022.884316). The above article is not cited in the present manuscript. Also, the authors may clearly distinguish the new work from the previous work as both works are identical.
The conclusion lacks the most relevant findings of the study.
The citing of articles in the body of the manuscript is not properly done and "Error! 41 Reference source not found" repeatedly appeared in the manuscript.
Response: Thank you very much for your comments on the manuscript. We have accepted all comments and made revisions. And the modified part is marked red. The manuscript has been polished by a professional polishing agency.
In the previous paper (https://doi.org/10.3389/fbioe.2022.884316), four nozzles were optimized by orthogonal method, and the parameter values were used to optimize the nozzle. Nanofibers were produced using four types of nozzles through centrifugal spinning experiments. The nanofibers fabricated by four nozzles were observed by SEM. The optimal combination of four nozzle parameters was determined. The results showed the conical-straight nozzle and curved-tube nozzle could spin better nanofibers. Furthermore, the fiber diameter fabricated by conical-straight nozzle was minimal.
In this article, we focus on optimizing the bent-tube nozzle. First, the mathematical model of spinning solution motion is established. Then, the optimization function of the bent-tube nozzle is obtained. Third, the neural network algorithm is used to optimize the structural parameters. The results reveal that when the bending angle is 15°, the curvature radius is 10mm, and the outlet radius is 0.205mm, the head loss of solution can be minimized. Final, centrifugal spinning experiments are conducted, and the influence of parameters of the bent-tube nozzle on nanofibers are analyzed. The results reveal that the optimized bent-tube nozzle improves the surface morphology of the nanofibers as their diameter distribution is more uniform.
We mainly focus on parameter combination optimization of bent-tube nozzles. When using optimized bent-tube nozzles to produce nanofibers, the head loss can be minimized. Then, by analyzing the influence of the bending angle, curvature radius, and outlet diameter of the bent-tube nozzle on the quality of nanofibers, we conclude that the optimized bent-tube nozzle produces nanofibers with better surface quality, smaller average diameter, and more uniform diameter distribution.
We have made corrections as required regarding the citation of references.
Comments on the Quality of English Language:
The language is slightly confusing and may need revision.
Response 1: The manuscript has been polished by a professional polishing agency, and the Certificate of Language Editing is attached.

Round 2
Reviewer 2 Report (New Reviewer)
Comments and Suggestions for Authors
The manuscript is revised thoroughly. It can accepted in the present form.
This manuscript is a resubmission of an earlier submission. The following is a list of the peer review reports and author responses from that submission.
Round 1
Reviewer 1 Report
Comments and Suggestions for Authors
Nanomaterials review: Fabrication and Optimization Mechanism of Nanofiber by Composite Spinning
This paper discusses the parameter optimization for centrifugal spinning of PEO/PVA polymer blend solution in water. The author modeled the nozzle flow and changed the bend angle of the nozzle to try to minimize the heat loss, then performed the centrifugal spinning experiment to form nanofibers.
Unfortunately, there are a number of shortcomings in this study that would undermine the claims made in this manuscript:
- The first paragraph of the introduction does not correspond to what the authors are studying. Quantum size effect has nothing to do with centrifugal spinning of polymers. The second sentence is incoherent. The authors need to clarify what field your study is in, then introduce relevant literature.
- There are several inconsistencies in the modeling.
o At line 198, the authors claim that k is related to the magnitude of φ. This “k” is different from the k that was defined in line 134. However, the dependence is not described anywhere in the text, nor is it explicitly calculated in deriving line 269.
o Line 228 does not make sense. If θ=90°, would the flow velocity be 0? This does not occur for incompressible fluids.
o Lines 197 and 241 show up without any references. Where did these coefficients come from? It also uses θ in radians rather than degrees as used everywhere else.
o In the case where θ=0°, hIV should be reduced to similar expressions as hI and hIII. This is not the case.
- In Line 301 the authors claim that the structural parameters are non-linear and non-monotonic for the head loss. While it is non-linear, it is monotonic as I see no local minima in Figure 5. This obviates the need for use of NN.
- Lines 322-323 does not specify why this was the case. Where did these numbers come from?
- Why is optimizing for head loss relevant? If reducing the frictional heat is the only variable to adjust for, keeping the opening wide and not angled would be the optimum. Certainly there are constraints such as forming stable jets, spinner to collector distance, etc. None of these are discussed in this paper.
- At the end of the paper, the authors perform exactly two experiments, one with 15° bend and one with straight fibers, which is far from proving that the optimization has indeed been successful. While the SEM image analysis seems to show a modest change in fiber diameter, the authors did not specify why this was related to the head loss they claimed to have optimized for.
- The TGA analysis shows that mass loss is different between the two samples, but the way the start/end points are taken are unclear. The magenta line seems to show just about the same mass loss as the green line if the magenta line base is taken at the plateau rather than the inflection point, in a similar fashion to the green line.
Given these shortcomings I do not recommend publishing this work.
Comments on the Quality of English LanguageThere are incoherent sentences throughout the manuscript. The authors should consult with professional proofreading services if they wish to publish this work in the future.
Reviewer 2 Report
Comments and Suggestions for Authors
It seems to me that the title of the submitted manuscript should be changed. At the moment it is not entirely clear and may be misleading to unprepared readers.
In the abstract of the manuscript, the authors indicate the topic of work, the main directions on which the authors concentrate, namely the production of nanofibers with improved surface morphology and a more uniform diameter distribution, which can be achieved by changing the geometry of the nozzle.
The introduction contains the main points devoted to nanofibers, namely their demand by humans in medicine, energy, production of composites, etc. Considering various methods of forming fibers, the authors do not note such important issues as solvent regeneration, the maximum concentration of the polymer in the spinning solution, the role and type of precipitants, toxicity of the solvents used.
After the theoretical part, the authors describe the proposed fiber spinning plant. Its essence is to use high rotation speeds to overcome viscous and surface forces and form small diameter fibers. The authors describe in detail the nozzle used and provide its corresponding drawings. In their calculations, the authors come to the conclusion that the head loss decreases with the outlet radius. The most suitable parameters, according to the authors, are bending angle 15°, radius of curvature 10mm, outlet radius 0.205mm. After identifying these parameters, the authors use them in practice. Mixed solutions of PEO and PVA are used as spinning systems. Solutions are formed through dies with a diameter of 0.205 and 0.410 mm. The quality of the obtained fibers (their average diameters) is assessed using SEM. It has been shown that for a spinneret with a smaller hole diameter, fibers with a smaller average diameter are formed.
I recommend moving Figure 3 closer to the place where it was first mentioned.
Line 134. “n is the rheological index” is probably a typo; the formula does not use such a designation.
In Table 1 you need to add dimensions.
Lines 53-55. I recommend changing this statement.
Figure 1. front and top views do not match in meaning, also the designations in the figure need to be corrected (3)
Line 359. The roughness of the fibers is difficult to assess from the presented micrographs. Therefore, it is necessary either to provide photographs with high magnification or to remove this argument from the manuscript.
Section 4.2. "Thermal decomposition experiment of nanofiber membrane", in my opinion, needs to be completely redone. It is advisable to add DSC and DTG data. Also, interpretation of thermal properties without structural data is difficult. The authors must explain what caused the change in the thermal behavior of the spun samples!
The presented manuscript is interesting, but at the moment it appears unfinished. The results of the experiment are left without in-depth analysis by the authors. It is better to include a description of the materials used in a separate section, indicating their detailed characteristics. I recommend that authors take into account the above comments to improve the quality of their work. I also recommend checking the English language (translation) of the manuscript; it may be better to involve a qualified editor for this.
Comments on the Quality of English LanguageEditorial work required with text.